# Lung Ultrasound in Critical Care: A Narrative Review

**DOI:** 10.3390/diagnostics15060755

**Published:** 2025-03-17

**Authors:** Lee Berry, Lucas Rehnberg, Paul Groves, Martin Knight, Michael Stewart, Ahilanandan Dushianthan

**Affiliations:** 1General Intensive Care Unit, University Hospital Southampton NHS Foundation Trust, Southampton SO16 6YD, UK; lucas.rehnberg@uhs.nhs.uk (L.R.); paul.groves@uhs.nhs.uk (P.G.); martin.knight@uhs.nhs.uk (M.K.); michael.stewart@uhs.nhs.uk (M.S.); 2School of Health Sciences, University of Southampton, Southampton SO16 6YD, UK; 3Shackleton Department of Anaesthetics, University Hospital Southampton, Southampton SO16 6YD, UK; 4Perioperative and Critical Care Theme, NIHR Biomedical Centre, University Hospital Southampton NHS Foundation Trust, Southampton SO16 6YD, UK; 5Clinical and Experimental Sciences, Faculty of Medicine, University of Southampton, Southampton SO16 6YD, UK

**Keywords:** lung ultrasound, point-of-care ultrasound, lung aeration, acute respiratory failure, COVID-19, weaning

## Abstract

Lung ultrasound (LUS) has become a crucial part of the investigative tools available in the management of critically ill patients, both within the intensive care unit setting and in prehospital medicine. The increase in its application, in part driven by the COVID-19 pandemic, along with the easy access and use of mobile and handheld devices, allows for immediate access to information, reducing the need for other radiological investigations. LUS allows for the rapid and accurate diagnosis and grading of respiratory pathology, optimisation of ventilation, assessment of weaning, and monitoring of the efficacy of surfactant therapies. This, however, must occur within the framework of accreditation to ensure patient safety and prevent misinterpretation and misdiagnosis. This narrative review aims to outline the current uses of LUS within the context of published protocols, associated pathologies, LUS scoring systems, and their applications, whilst exploring more novel uses.

## 1. Introduction

The use of lung ultrasound (LUS) at the bedside in critical care has accelerated over the last decade since Lichtenstein’s work on the BLUE protocol was published [1], which described the interpretation of LUS images and their application to a clinical protocol decision tree to differentiate the causes of acute respiratory failure. The COVID-19 pandemic further increased and accelerated interest in LUS as a clinical tool to aid direct bedside patient care, with a large number of published works providing new data on urgent clinical applications [2].

The adoption of mobile ultrasound machines and handheld probes offers the immediate, versatile assessment of patients both within the intensive care unit (ICU) and in the prehospital setting. Traditionally, these critically ill patients would have required portable radiological imaging or transfer to radiology departments for investigations. The availability of mobile ultrasound has reduced radiation exposure, saved time, and reduced capacity, thereby minimising additional risks associated with the internal transfer of critically ill patients [3]. The accuracy of LUS compared to traditional imaging for pleural abnormalities, pneumothorax, pulmonary oedema, and pneumonia has been thoroughly investigated, with ultrasound consistently matching or exceeding the capabilities of a chest radiograph for diagnosis and correlating with Computerised Tomography (CT) findings [4,5,6,7]. The understanding of lung ultrasound use in critical care has significantly evolved. The previous belief that air impedance prevents reliable interpretation has changed, and lung ultrasound (LUS) is recognised as an important tool in patient care, complementing traditional physical examinations [8,9].

This rapid integration of POCUS and LUS may present potential risks, including the possibility of misinterpretation when undertaken without formal training and qualification [10], and it requires a high level of skill and experience to interpret. While there is no international consensus on the appropriate level of training required for non-radiologists, several societies and professional bodies have sought to standardise training and assessment. However, the variation in training and the number of scans required may differ. In the UK, bodies such as the Intensive Care Society (ICS) with its Focused Ultra-Sound in Intensive Care (FUSIC) modules and the Royal College of Emergency Medicine (RCEM) curriculum aim to provide a standard for training and assessment. The British Thoracic Society (BTS) provides additional guidance on training standards for respiratory specialists, primarily concentrating on pleural pathologies [11].

Within our own critical care unit, we routinely use LUS for clinical diagnosis and recognise the importance of LUS and wider point-of-care ultrasound applications in patient care. Consequently, this narrative review aims to discuss the current understanding and clinical applications of LUS at the bedside, as well as its potential future directions. We screened for relevant articles on PubMed, CINAHL, and Google Scholar using medical subject headings, keywords, and synonyms related to ultrasonography (e.g., ultrasound, LUS, point-of-care), lung (e.g., thorax, pulmonary), and lung diseases (e.g., pneumothorax, effusion, consolidation, atelectasis, pneumonia, acute respiratory distress syndrome, oedema) until July 2024.

## 2. Lung Ultrasound

### 2.1. Ultrasound Physics Basics

The generation of an ultrasound image is produced by a transducer probe emitting high-frequency sound waves in the range of 2 to 15 million hertz (Mhz). This high-frequency sound wave allows for the real-time, non-invasive acquisition of images. The ultrasound transducer emits acoustic energy via the piezoelectric effect, where electrical energy pulsed across crystals causes them to oscillate rapidly, producing a repeated wave sequence of compressions and troughs [12]. These waves can be reflected by tissue interfaces with varying acoustic impedance throughout the viewing field and returned to the crystal, causing oscillations and thereby generating an electrical signal to construct an image [13]. A group of piezoelectric crystals, an array, is cyclically activated in phase, thereby producing an arc of ultrasound lines that results in a 2D image.

The formation of the image is dependent upon the interaction of the ultrasound with tissue; as such, the ultrasound may be reflected, refracted, scattered, or absorbed as it passes through varying structures such as air, soft tissue, fluid, or bone [14]. As a result, the image produced from this attenuation can be interpreted.

Available ultrasound machines may have three different transducer probes: linear, curvilinear, and phased array. No specific probe is indicated for LUS, and each offers advantages due to the variation in frequency and depending on the clinical question posed [15]. The high-frequency (6–15 Mhz) linear probe may be used for superficial structures such as the pleura; the curvilinear probe (<6 Mhz) is recommended as the single best probe to elicit all LUS signs, providing the best resolution; and the phased array probe (1.5–7.5 Mhz), with its small footprint, allows for easy access between ribs and identification of deeper structures, although its resolution is less than that of both the linear and curvilinear probes [12,15]. While air is a poor conductor of sound waves, the pleural line of aerated lung and the subsequent changes due to pathological processes lead to artefacts that aid in diagnostic interpretations [1,6,16].

### 2.2. Lung Ultrasound Examination

The LUS examination can be performed with the patient in a semi-recumbent supine position, hence its attractiveness as an adjunct to clinical care in critical care units when patient movement may be restricted. Conducting the examination requires an understanding of surface anatomy and its relation to probe positioning. The LUS examination and the scanning protocol are reliant upon this surface anatomy and can be both simple and complex. A simple examination may involve scanning three regions of each lung and interpreting these images, and a more complex examination may incorporate 12–28 regions of the lung fields whilst applying a grading score to indicate pathology [6,17,18,19].

The BLUE protocol is one such basic LUS protocol for undifferentiated patients with respiratory failure; it involves accessing three standardised points on each hemithorax and assessing both the upper anterior and lower anterior chest, as well as the Posterior Lateral or Pleural Syndromes (PLAPS) point. The images are analysed to establish the presence or absence of lung sliding, A-lines, B-lines and consolidation, thus allowing for the identification of the most significant pathologies via a decision tree. When performing any LUS, the probe should be in a longitudinal position (cephalad–caudad) with a depth of 6–10 cm, where artefacts form from the pleural line and can be examined. Lung presets may be available on the USS machine, which can ensure that image settings and processing are optimised for interpretation.

### 2.3. Lung Ultrasound Artefacts and Their Normal Findings in an Aerated Lung

The crucial diagnostic signs in LUS are related to the artefacts generated by the subcutaneous tissues and the air beneath. Beginning with Brightness mode (B-mode), the normal findings are demonstrated in Figure 1. With the lung centred in the middle of the image, the soft tissues and ribs can be seen, with the bright white pleural line located approximately 0.5 cm below. The ‘bat sign’ is useful analogy to help identify the correct probe position, as the ribs cast a shadow at the bottom of the image, with the pleura identifiable in between.

The parietal and visceral pleura can be seen as a bright white line between the rib spaces. This pleural line can be seen to be sliding with tidal ventilation, creating a shimmering movement artefact that is suggested to be reminiscent of ants walking on a tree branch. This line represents the interface between the soft tissues of the chest wall and the aerated lung beyond. The pleura itself is thin and smooth in a normal aerated lung, and changes to this pleural line indicate potential pathology [2,20].

Below the pleural line and within normal aerated lung, a ring-down artefact termed A-lines can be visualised. These artefacts emanate from the pleura itself when it is thin and dry, allowing the observer to infer its existence. The pleura reflects some of the ultrasound beam back towards the transducer, and this, in turn, is reflected back to the pleura. Once reflected to the probe, a line is produced at the same distance below the pleura as the probe is above it. This artefact is often seen repeated below the pleural line at multiples of the probe–pleural distance, as sound waves are reflected back and forth, and signifies air. The presence of A-line artefacts, coupled with pleural sliding, indicates normal aerated lung [1].

B-line artefacts are comet tail-like beams that originate at the pleural line and ‘shine’ down to the bottom of the screen. They are hyperechoic and well-defined and move backwards and forwards with tidal ventilation, thereby obliterating A-lines as they pass [6]. The physics of B-lines is not entirely understood, but their formation may be caused by areas of subpleural interlobular oedema. Up to 2–3 B-lines may be seen in healthy individuals, particularly at the bases of the lungs. However, three or more B-lines between the ribs are considered pathological. As the pathological processes increase, these B-lines may coalesce, leading to a white lung appearance with total obliteration of A-lines.

E-line and Z-line artefacts are referred to as false B-lines and are often mistaken for them, but they may be present in a normal LUS. E-lines are vertical lines caused by gas trapped within the subcutaneous tissue. They do not move with respiration nor arise from the pleural line; Z-lines are bundle-shaped vertical lines that are ill-defined, do not move with ventilation, and do not erase A-lines [21].

### 2.4. Altered Lung Ultrasound Artefacts in Critical Care Pathology

More than three B-lines are also termed ‘lung rockets’ and are considered pathological (Figure 2A). B-lines that coalesce and occupy more than 50% of the rib space are termed ’ground glass rockets’ [6]. On CT scans, B-lines often correlate with thickened intralobular septa and subpleural consolidation. The pathologies that can create B-lines are mainly divided into those that increase intralobular septal size and those that affect the pulmonary interstitial tissue. The former clinically corresponds to pulmonary oedema (whether cardiac or neurogenic), and the latter pertains to interstitial disease. The presence of B-lines has been associated with a Pulmonary Capillary Wedge Pressure (PCWP) of greater than 18 mmHg [22], although this is only the case if an intralobular disease process is occurring.

B-lines have specific features that when combined with full LUS imaging and interpretation, can help differentiate between cardiogenic pulmonary oedema, Acute Respiratory Distress Syndrome (ARDS), interstitial pneumonia, and fibrosis (Table 1) [23].

The absence of pleural sliding has been used to diagnose pneumothorax, as this absence could suggest separation of the pleura [5]. Lung sliding can also be assessed with M-mode. The normal appearance resembles a ‘seashore’, with a contrast between the superficial muscles and parietal pleura and the deeper (moving) lung tissue. In pneumothorax, there is no movement; thus, a ‘barcode or stratosphere’ sign is seen, with no obvious delineation between the lung and superficial structures [6,24]. The abrupt loss of lung sliding adjacent to a section of sliding lung is called the ‘lung point’. The identification of the lung point is highly sensitive and specific for pneumothorax [16,24].

It must be noted that there are a number of circumstances in which there is no lung sliding and no pneumothorax. Hyperinflated lungs in emphysematous COPD patients, bronchospasm with gas trapping, densely consolidated lung, and endobronchial intubation are examples that involve a lack of lung movement on US that is not due to pneumothorax. A useful sign to differentiate these causes from pneumothorax is the ‘lung pulse’. This refers to the transmission of the ‘pulse’ or pressure wave through the lung parenchyma when the heart beats, indicating that the lung is inflated from the heart to that point on the thoracic wall. If the lung pulse is absent, there is no lung in contact with the thoracic wall; in conjunction with absent lung sliding, the ‘barcode’ sign is highly suggestive of a pneumothorax, even without a ‘lung point’ [6,25]. The shred sign is seen when the smooth, regular pleura is replaced by a thickened or ragged pleura, which is associated with an irregular border [26]. This sign is associated with subpleural consolidation and, on CT, is often shown as consolidation that reaches the level of the pleura. Areas of shred can sometimes be accompanied by E-lines, which originate from the area of consolidation and travel vertically in a similar fashion to B-lines. Consolidation on an ultrasound is easily seen and has the appearance of dense heterogeneous tissue (Figure 2B). Distinguishing between (infective) consolidation and collapse can be difficult. Air bronchograms are a sensitive sign of consolidation related to infective processes, mirroring the appearance on a radiograph. However, if these bronchograms are static, it is more likely that an atelectatic process is occurring rather than infective one. The presence of high blood flow through the consolidated lung may indicate an infective process, while the absence of blood flow could indicate simple collapse or pulmonary infarction.

Pleural effusions are well demonstrated on lung ultrasound, outperforming chest radiographs, with a high sensitivity of 93% vs. 43% [1,4,27]. It can also detect much smaller volumes of effusion compared to a chest radiograph.

While the definitive differentiation of whether the fluid is likely transudate or exudate requires pleural fluid analysis, LUS can help distinguish between a simple and complex pleural effusion. Complex effusions are associated with the presence of fibrin deposits, septations, or hyperechoic fluid. Complex effusions are often, but not exclusively, exudates. Figure 3A shows a complex effusion with fibrin stranding, likely exudative in nature. This may also guide decision-making regarding drain types (e.g., small or large bore Seldinger vs. surgical) and whether surgical intervention is required. LUS also helps estimate the quantity of pleural effusion present, which is comparable to the volume drained [28,29]. Large effusions often have free-floating lung tissue in them, creating the ‘jellyfish sign’. Other pathological processes, such as haemothorax (Figure 3B) and abscess, can also be elucidated with experience. The British Thoracic Society, UK, now recommends that pleural procedures only be undertaken with LUS guidance [30].

## 3. Disease-Specific LUS Use

### 3.1. Adult Respiratory Distress Syndrome (ARDS) and LUS Findings in ARDS

ARDS is a life-threatening heterogeneous respiratory syndrome that applies to a spectrum of conditions with differing etiologies, which share common pathological characteristics [31,32]. It is defined by the Berlin Diagnostic Criteria as an acute, diffuse inflammatory lung injury precipitated by risk factors such as pneumonia, non-pulmonary infection, trauma, transfusion, burns, aspiration, or shock; this condition leads to non-hydrostatic pulmonary oedema, gravity-dependent atelectasis, and the loss of aerated lung tissue [31,32,33,34]. Whilst accounting for 23% of ventilated intensive care patients, it has an estimated mortality rate of 45% [32].

In patients with ARDS, LUS follows a pattern of aeration to de-aeration and increased lung density secondary to non-cardiogenic pulmonary oedema. This results in a change on a continuum from normal lung sliding with preserved A-lines to pleural thickening with irregularities, B-line formation with an increase in number and distribution, subpleural consolidations (shred sign), and areas of complete consolidation as the disease moves from aeration to complete de-aeration [16,35]. The new global definition of ARDS [33], which builds upon the Berlin definition [36], suggests that LUS should be recognised as a modality to identify the loss of lung aeration (B-line formation), which is consistent with non-cardiogenic pulmonary oedema, with or without consolidation, by appropriately trained staff in lieu of CXR or CT, particularly in resource-limited settings to aid in ARDS diagnosis [33].

#### LUS Scoring in ARDS

A number of LUS scoring systems [37,38,39,40] have been proposed to provide a quantitative approach to lung aeration assessment and have been shown to strongly correlate with lung density measured by quantitative CT [41,42]. A score can be derived from 12 regions of the lung (6 per hemithorax), with a global score resulting from the sum of the scores assigned to each field. Each field is assessed for four steps of progressive loss of lung aeration [40]. This scoring system proposes the quantification of the extent to which B-line artefacts occupy the pleura to assess loss of aeration (Table 2).

The quantification of B-lines in LUS as a surrogate for increased extravascular lung water (EVLW) has demonstrated a linear correlation with measurements derived from transpulmonary thermodilution [38]. Therefore, with homogeneous loss of aeration, as seen in cardiogenic oedema, B-lines become more and more prevalent, finally coalescing and involving the whole pleura. The non-homogeneous nature of ARDS demonstrates non-focal areas of coalescence, with variable changes in the percentage of pleural involvement [40]. Changes in the LUS in ARDS patients receiving fluid resuscitation were the earliest parameters suggestive of increased EVLW, which preceded deleterious effects on gas exchange [39].

This LUS score has also been employed to monitor reaeration after antibiotic therapy for ventilator-associated pneumonia and during recovery from ARDS in patients receiving Extracorporeal Membrane Oxygenation (ECMO) by demonstrating improvements in the cumulative scores [42].

A further refinement of this score has been proposed. The LUS-ARDS score [43,44] incorporates the assessment of an aeration score within a standard 12-lung region protocol and further assessment of 8 anterolateral lung regions for pleural abnormalities (such as abnormal pleural line, subpleural consolidations, dynamic air bronchograms, and pleural effusion) with a score ranging from 0 to 91. This LUS-ARDS score demonstrated high accuracy in identifying ARDS with a high score (>27) or excluding it with a low score (<8). Intermediate LUS-ARDS scores may indicate the probability of ARDS being present. Therefore, the LUS-ARDS scoring system provides clinicians with an estimate of the probability of ARDS that is consistent and fulfils the criteria of the current Berlin definition of ARDS.

### 3.2. LUS Use in COVID-19

The COVID-19 pandemic presented an international health crisis that placed significant pressure on healthcare systems within intensive care settings. The wide range of presentations and severity of the disease required prompt diagnosis, treatment, and risk stratification. LUS was proposed as a method to aid in diagnosis, triage, and clinical management, being attractive due to its low cost, comparative ease of use, and rapid access compared to more traditional radiological techniques. COVID-19 has a predictable disease progression in the lung parenchyma, ranging from mild to severe disease and progressing from the distal regions of the lung with alveolar damage, interstitial thickening, and consolidation [45]. These pathological changes lead to distinctive LUS findings as the disease progresses from pre-disease (normal LUS scan of aerated lung) to critical illness (significant LUS abnormalities, completely de-aerated), similar to the findings in ARDS [45,46,47] (Figure 4).

Although COVID-19 has some distinctive features on LUS such as ‘moth eaten’ irregular pleura and B-lines, no single LUS finding is pathognomonic for COVID-19 and may be seen in other disease processes, as outlined. Therefore, LUS findings in COVID pneumonia, viral pneumonia/pneumonitis, and ARDS may overlap [48], and clinical correlation is often required. Due to the non-homogeneous nature of COVID-19 in the lung parenchyma, disease progression can be monitored over time, from confirmed infection to mild or severe illness, and finally to recovery.

The gold standard COVID-19 diagnostic test remains the quantitative polymerase chain reaction (qPCR). However, the sensitivity of qPCR decreases from 90% at the onset of symptoms up to day 5 to 70% on days 9–11 [49] and typically can take several hours to return a result. Presentation to healthcare services with symptoms ranges from 3 to 10 days [50]; thus, the sensitivity of the qPCR may be much lower than anticipated on presentation. Radiological modalities, such as chest radiographs, may be suggestive of COVID-19, but the range of changes or their absence may result in diagnostic uncertainty, with approximately 40% of positive cases being missed [46]. Whilst CT scan findings of ground glass opacities in the peripheral and lower lung distribution can be suggestive of COVID-19 pneumonitis, this method is not available in all clinical settings. In addition, patients may exhibit no specific distribution, the findings are dependent upon the day of presentation and clinical context, and such ground glass appearances can be present in alternative lung pathologies [51].

Although CT has a high sensitivity for the diagnosis of COVID-19, it is impractical to use as a diagnostic test during pandemic surges for all patients presenting with nonspecific respiratory symptoms due to healthcare costs, radiological exposure, and the transportation of critically ill patients. The high level of agreement between LUS and CT scans in diagnosing interstitial pneumonitis secondary to COVID-19 has been demonstrated, and therefore, LUS could be considered an equivalent alternative where CT is not available [3]. The attractiveness of point-of-care LUS, which is known to be more sensitive than CXR in the diagnosis of interstitial patterns, and the unique ultrasound findings present in viral pneumonia can improve diagnostic accuracy, especially in populations with a high prevalence.

#### LUS Scoring in COVID-19

The severity of COVID-19 and the accompanying LUS characteristics have been used to develop LUS scoring systems to aid in diagnosis and risk stratification. A LUS scoring system was designed to assess the likelihood of a COVID-19 diagnosis at the point of admission to the hospital [52]. The stratification includes low probability (LowLUS), high probability (HighLUS), intermediate probability (Int LUS), and alternative (AltLus) probability, depending on the findings [52]. The high LUS probability is classified as coalescent B-lines, irregular pleural lines, and multifocal consolidations, whilst intermediate LUS is classified as a unilateral B-line pattern with or without peripheral consolidations. HighLUS and IntLUS demonstrated a combined sensitivity of 90.2% in identifying patients with positive PCR.

An alternative scoring system [53] for patients with COVID-19 involves an acquisition protocol of 14 areas comprising 3 posterior, 2 lateral, and 2 anterior regions. This protocol was shown to identify patients with a higher rate of pathological lung areas (higher score), which were associated with worsening disease, increased risk of intensive care admission, and death [54,55]. A similar scoring system composed of 12 pulmonary zones also demonstrated an increasing score over time in non-survivors and in patients who developed Ventilatory Acquired Pneumonia (VAP) [18].

For ventilated patients, a phenotype stratification algorithm was proposed to aid in ventilation strategies to differentiate between low-elastance (Type L) and high-elastance (Type H) patients [56]. The Type L or COVID-19 ‘happy hypoxic’ classical presentation suggests a LUS that may appear normal or show a broken pleural line with accompanying B-lines. In contrast, the Type H patient would present with severe disease and subsequent subpleural consolidation on LUS. As compliance is the inverse of elastance, ventilator management for these two differing phenotypes needs to be tailored to the respiratory mechanics present. The serial use of LUS thereby allows for the diagnostic and daily monitoring of lung aeration in order to help predict responses to prone positioning and PEEP strategies [56].

A meta-analysis comparison of LUS and CT in the diagnosis of interstitial pneumonia by Wang et al. [3] demonstrated that the agreement of LUS compared to CT is high, and therefore, LUS could be considered an equally accurate alternative to CT when tests are delayed or not available. A systematic review by Lai et al. [57] studied the correlation among the partial pressure of oxygen to the fraction of inspiratory oxygen (PO_2_/FiO_2_), intubation rates, and mortality in relation to LUS scoring systems. A higher LUS score was strongly correlated with worsening PO_2_/FiO_2_ ratios and increasing intubation rates; it was higher in the intubated patient group, reflecting the severity of the disease, whilst a lower LUS score was found in survivors.

## 4. LUS and Mechanical Ventilation

### 4.1. LUS and Ventilation Strategies

Focal versus diffuse patterns of ARDS allow for an individualised approach to ventilation that has been associated with improved outcomes. Proning is considered beneficial when there is a loss of aeration in predominantly infero-posterior areas and normal aeration in the anterior-apical areas. Conversely, when there is a loss of aeration and lung ultrasound scores are high in a diffuse pattern, a high PEEP strategy can be associated with benefits [44]. The identification of atelectatic lung by LUS can also guide recruitment manoeuvres and PEEP titration. The most dependent zone of the atelectatic lung is scanned while performing a recruitment manoeuvre, looking for progressive lung reaeration to define the lung’s opening pressure. The closing pressure is acquired during a PEEP recruitment trial using the same principles. Echocardiography is also useful for assessing haemodynamic stability prior to performing such recruitment manoeuvres [58].

### 4.2. Weaning

Tests to ascertain suitability for extubation, such as a Spontaneous Breathing Trial (SBT), can often stress a patient’s neurological, respiratory, and cardiovascular systems without providing information on causality in the event of a trial failure. The transition from a positive pressure mandatory mode of ventilation to spontaneous breathing with negative intrathoracic pressure breaths is associated with increased venous return and loading of the right ventricle. The negative pressure breaths and the elevated adrenergic tone that often accompany the reduction in sedation can also increase left ventricular afterload [59]. A recent meta-analysis has demonstrated that diastolic dysfunction is more useful than systolic dysfunction in predicting weaning failure [60]. Diastolic dysfunction, exacerbated by the haemodynamic changes associated with weaning, predisposes individuals to pulmonary oedema. This weaning-induced pulmonary oedema results in increased EVLW that is homogeneously distributed within the lung parenchyma, manifesting as B-lines. Increased B-lines during SBTs correlate with weaning-induced pulmonary oedema and weaning failure [61].

The existing loss of aeration or decruitment during a SBT can be assessed and monitored using the LUS score. Elevated LUS scores predict SBT failure, and in patients who pass their SBT, a higher LUS score predicts post-extubation respiratory distress [62].

Diaphragmatic dysfunction, whether pre-existing, traumatic, or acquired through critical illness neuromyopathy or other neuromuscular weakness, can be associated with weaning failure. There are different methods for assessing diaphragmatic function using lung ultrasound. The first method assesses diaphragmatic excursion, with an excursion of <25 mm being bilaterally suggestive of severe dysfunction and predictive of weaning failure [63]. However, diaphragm displacement is highly variable in mechanically ventilated patients due to different levels of PEEP and positive pressure, which can mask a paralysed diaphragm. The second approach is to measure diaphragm thickness, or more specifically, the diaphragm thickening fraction (thickening fraction = (end-inspiration thickness − end-expiration thickness)/end-expiration thickness). The lower the thickening fraction, the higher the probability of weaning failure. However, the optimal thickening fraction threshold to predict successful weaning varies from 20 to 36%, depending on the different approaches to weaning employed in the different weaning trials [58].

An integrated thoracic ultrasound evaluation encompassing lung, diaphragm, and cardiac sonographic data has been demonstrated to accurately predict post-extubation distress in a patient who passes an SBT, with a receiver operator characteristics (ROC) area under the curve (AUC) of 0.972 at the start and 0.92 at the end of the SBT. Notably, this evaluation includes the identification of lung bronchoconstriction or upper airway oedema and secretions. However, the abovementioned techniques aid the clinician in risk stratifying patients according to the probability of success and provide diagnostic information to guide treatment. In optimising weaning, both interstitial oedema and surrogates for left ventricular diastolic dysfunction were the best predictors [64].

However, there are numerous pathologies, including extra-thoracic causes such as neurological diseases and ventilatory demands, including increased resistive load, that are not easily assessed by LUS protocols.

## 5. Limitations

LUS, similar to other imaging modalities, is operator-dependent for image acquisition and interpretation [26], with misdiagnoses of lung pathologies, such as pneumothorax and consolidation, being reported in the literature [65]. Interrater reliability for common findings, such as pneumothorax, pleural effusions, and B-lines, has been shown to be high after short training sessions comprising a few hours [26,66,67]. However, the artefacts presented may be difficult to interpret and may have competing clinical causes that require training and continued exposure. A four-step level of knowledge has been suggested: (1) the basic simple identification of normal and consolidation; (2) an intermediate level involving pleural involvement and pleural effusion; (3) an advanced level including LUS scores as a monitoring tool; and (4) an expert level including qualitative and quantitative LUS in the clinical management of respiratory failure [15].

As LUS is now considered a key component of a clinical examination and an extension of the traditional physical examination, both scientific societies and educational curricula have attempted to quantify the nature and length of training required [68]. Training periods to acquire skills from 25 to 40 examinations have been suggested [9,69], although it may be unrealistic to ensure that trainees encounter all relevant pathology during this limited exposure [70]. Further research is required to understand the educational requirements for LUS and its implementation, from basic to advanced levels [15,68].

There are also limitations beyond the operator’s education and experience that lie with the patient. A large body habitus can impair visualisation due to the thickness of the chest wall, whilst surgical emphysema precludes the propagation of ultrasound beams. Surgical dressings prevent image acquisition, whilst mechanical ventilation and patient position may hinder access to the required lung regions and appropriate image acquisitions [71].

## 6. Future Directions

### 6.1. Surfactant Therapy in ARDS, COVID-19, and Neonatal RDS

ARDS due to alveolar damage and acute inflammation results in alveolar and endothelial injury, leading to capillary leak, pulmonary oedema, collapse, and consolidation [31,32,72]. Surfactant abnormalities occur with reduced surfactant synthesis, secretion, and recycling, as well as increased surfactant breakdown and increased surfactant inhibition [72]. Pulmonary surfactant is produced by type II pulmonary epithelial cells and reduces alveolar surface tension, preventing alveolar collapse and maintaining immune response in health [73]. Pulmonary surfactant as a therapy is a standard treatment in neonates with respiratory distress syndrome (RDS) and has been an area of investigation in ARDS for several decades. However, despite promising animal trials, it has failed to demonstrate an improvement in mortality in systematic reviews [73,74]. Whilst a Cochrane review was uncertain if any difference existed in early or late mortality, it also did not find a reduction in ventilator days or duration [75]. However, surfactant use during the COVID-19 pandemic in patients with pneumonitis demonstrated some promising early results, such as improvements in oxygenation, ICU length of stay, and the need for mechanical ventilation [76], although these were small-scale studies, which prevent wide-ranging adoption of the results, and they should be treated with caution. Therefore, due to the heterogeneous nature of ARDS, further large-scale research is required to determine what type of surfactant may be indicated, the dose, the delivery mode, the fate of the delivered surfactant, the proportion of surfactant that is surface-active in vivo, and how to reduce surfactant inhibition and breakdown [72]. LUS may help with future surfactant research through comprehensive LUS examinations and scoring systems that aid in the identification of non-aerated regions of the lung, possible targeted surfactant delivery and progression, and recovery from disease, as has been evidenced in neonatal RDS.

The early assessment and use of surfactant therapy in neonatal (RDS) have been shown to benefit from the integration of LUS into the treatment pathway [77]. Neonatal respiratory distress occurs in a large number of preterm babies, with incidence decreasing with increasing gestational age. While it affects only 0.3% of neonates born at ≥38 weeks gestation [78], that number increases to 93% in those born at ≤28 weeks [79]. RDS is associated with multiple acute complications, such as alveolar rupture, infection, and pulmonary haemorrhage, as well as conditions such as bronchopulmonary dysplasia (BPD) and neurological impairment. Early treatment with exogenous surfactant improves outcomes and is recommended by international guidelines [80]. The LUS score has been used to assess the severity of lung involvement in RDS and is highly predictive of the need for surfactant administration [81]. It was first described by Brat et al. [82], who divided each half of the chest into three areas: upper anterior, lower anterior, and lateral. Each region was scanned with a linear microprobe and given a score of 0–3 depending on the degree of pathology identified. A score of zero is given for an A-pattern (A-lines only), one point is given for a B-pattern (≥3 well-spaced B-lines), two points are given for a severe B-pattern (crowded or coalescent B-lines with or without consolidations limited to the subpleural space), and three points are given for extended consolidation. The maximum score of 18 represents severe lung pathology.

The use of LUS has been integrated into the recent European Guidelines, which recommend the administration of surfactant if lung ultrasound suggests surfactant deficiency, although how this is assessed is not specified [80]. A meta-analysis of 10 studies by Luo et al. in 2023 [83] found that a LUS score of 5 had high sensitivity and specificity for surfactant administration, which is consistent with a previous systematic review and meta-analysis [84]. Further work is required to determine the optimal score; however, using lung ultrasound to direct surfactant administration in neonatal RDS has been shown to result in earlier surfactant administration, decreased duration of mechanical ventilation, and reduced oxygen exposure [85,86]. The LUNG study, an international multicenter RCT that aims to assess the use of lung ultrasound on the rates of BPD and death, is currently recruiting patients and may provide further evidence to guide this field [87].

### 6.2. Artificial Intelligence (AI) in LUS

AI is being applied in medical imaging to improve diagnostic accuracy [68], and both Machine Learning (ML) and Deep Learning (DL) are being applied to LUS to aid in the interpretation of the captured artefact [88]. One of the limitations of LUS is the similarity of artefacts that are present across pathologies, such as in B-lines, as well as LUS being operator-dependent in image acquisition. Indeed, point-of-care machines, such as those produced by GE, have the ability to count B-lines and exhibit consistent intra-observer reliability [89]. DL models applied to B-line analysis have shown that a DL algorithm is able to perform at a high level of accuracy in identifying B-lines [90].

Since COVID-19, ARDS and cardiogenic pulmonary oedema have all demonstrated similar B-line artefacts, making clinical diagnosis challenging. A DL model has been applied to a dataset containing COVID-19, non-COVID, ARDS, and pulmonary oedema [91]. The DL model was able to outperform clinicians in distinguishing between the pathologies (*p* < 0.01) [91]. DL models have been applied to chest radiographs, CT scans, and LUS in the identification of COVID-19 [92]. The LUS DL model also outperformed the chest radiograph DL model and showed 100% sensitivity and 100% predictive value.

The utilisation of DL models in pneumothorax detection (absence/presence of lung sliding) has also demonstrated a sensitivity of 82–86% and a specificity of 76–92% [93,94]. The production of robust ML and DL applications requires a large-volume data set in order to train the algorithm; however present studies are small in scope, using images taken from different machines and operator parameters [68]; therefore, future research should be large-scale, with well controlled and consistent imaging parameters (machine, depth, focal point), so that AI can improve diagnostic accuracy in real time at the bedside to aid the clinician.

### 6.3. LUS in Phenotyping and Personalised Therapy for ARDS

Optimal ventilation management in ARDS is challenging due to the heterogeneity of the disease. Often, two competing phenotypes exist, the non-focal, with diffuse patchy loss of aeration, and the focal, in which predominant dorsal inferior consolidations are present [95,96]. These non-focal patients respond to higher levels of Positive End Expiratory Pressure (PEEP) and recruitment manoeuvres while focal patients respond to lower levels of PEEP and prone positioning. The personalised mechanical ventilation guided by ultrasound in patients with acute respiratory distress syndrome (PEGASUS) study [97] aims to undertake a 12-point LUS exam and score to differentiate between these two sub-phenotypes and provide two ventilation strategies in the personalised group (focal and non-focal) versus standard of care ventilation settings. The primary objective is to determine if personalised ventilation based on lung morphology assessed by LUS scores leads to a reduction in all-cause 90-day mortality, with a range of secondary objectives, such as reduced 28-day mortality, ventilatory-free days, and shorter length of stay [97]. This study may further enhance the diagnostic utility and application of LUS.

### 6.4. LUS in Prehospital and Emergency Medicine

The development of small handheld ultrasound devices has allowed for the assessment of undifferentiated patients by prehospital and emergency care practitioners, thereby to improving their diagnostic accuracy [98]. Prehospital POCUS use has been shown to have particular benefits in cases of cardiac arrest, to rule in or out cardiac activity; in trauma, to predict the need for interventions, such as pneumothorax, or the presence of free fluid within the abdomen; and in dyspnoea caused by congestive heart failure [99]. Considering point-of-care LUS alone in the assessment of undifferentiated patients with dyspnoea, paramedics experienced in the BLUE protocol and the BLUE profiles of pneumothorax, pneumonia, and pulmonary oedema have shown high diagnostic accuracy in the prehospital setting, in addition to aiding hospital diagnosis [100,101,102]. The recognition and treatment of acute heart failure by LUS in the prehospital setting has also been shown to improve paramedic diagnostic accuracy, and this early recognition allows for improved prehospital time to treatment [103]. Despite these potential benefits, prehospital POCUS has yet to demonstrate a mortality benefit for patients [104]. Concerns regarding the prehospital interpretation of LUS images could be offset by real-time image review and supervision by trained POCUS clinicians in emergency medicine [105]. The introduction of AI models that can be translated and easily used in prehospital settings may surpass the need for real-time supervision.

While the use of LUS in the prehospital setting is increasing, similar to its application in critical care, it is important to ensure that appropriate standards and training are established to provide the safe and effective use of lung ultrasound in the prehospital setting [98,99]. Currently, in the UK, POCUS is being used by most prehospital services. However, a major barrier to the adoption of POCUS is the lack of governance and literature supporting its use in the prehospital setting, despite the perceived benefits to patient care [104] Further prehospital LUS research should be based on diagnostic accuracy and the associated time to treat patients, and the potential for AI-aided interpretation should be explored.

## 7. Conclusions

LUS has been shown to be a key clinical application in the management of critically ill patients. While there are some established protocols, the scope of LUS usage is expanding. Its suitability for identifying pathology at the bedside was illustrated by its urgent adaptation during the COVID-19 pandemic. As the pandemic recedes, there is a need to reflect on its widespread use and the educational requirements for clinicians, both within traditional hospital critical care environments and in prehospital medicine, to ensure safe interpretation from basic to advanced levels. The development of specific protocols and scoring systems, as well as an appreciation of lung, pleural, and diaphragm ultrasound for weaning, tailored ventilation strategies, and other therapeutics, are the next steps that require further evaluation through clinical trials.

## Figures and Tables

**Figure 1 diagnostics-15-00755-f001:**
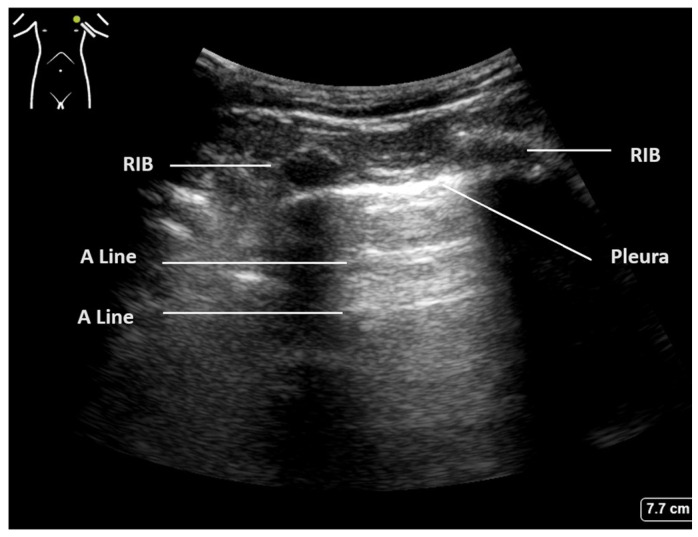
Normal lung ultrasound of aerated lung demonstrating ’bat wing’ appearance of ribs and A-lines.

**Figure 2 diagnostics-15-00755-f002:**
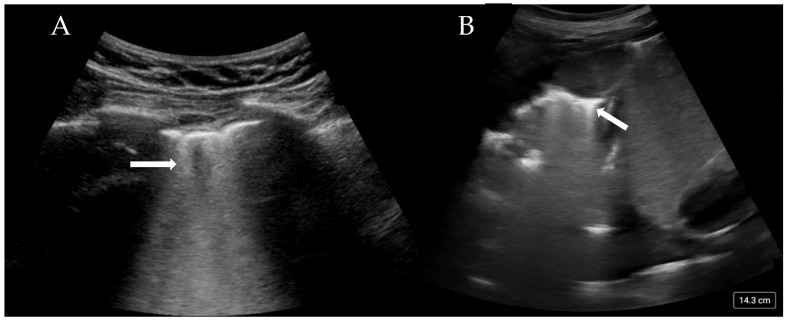
(**A**) Confluent B-lines (arrow) consistent with pulmonary oedema and (**B**) consolidated and collapsed lung with shred sign (arrow).

**Figure 3 diagnostics-15-00755-f003:**
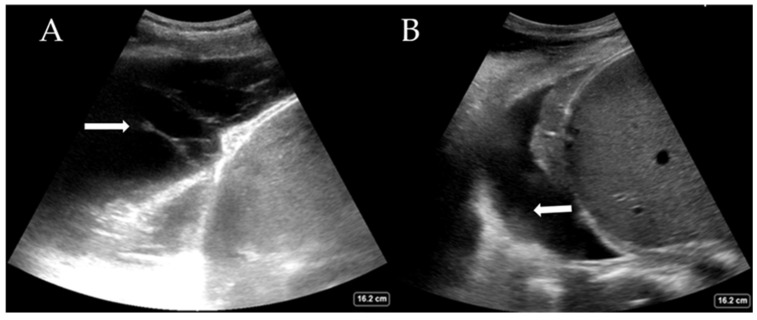
(**A**) Complex pleural effusion with fibrin stranding (arrow) and (**B**) haemothorax with adherent haematoma on the diaphragm (arrow).

**Figure 4 diagnostics-15-00755-f004:**
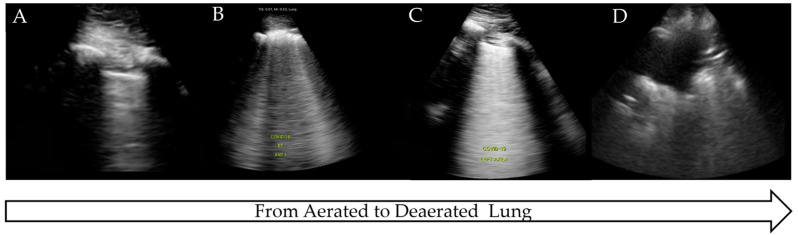
LUS findings on COVID-19 progression. (**A**) Normal aerated lung, (**B**) irregular pleural line with B-lines, (**C**) B-lines on >50% of image: ‘white lung’, (**D**) completely de-aerated collapsed lung.

**Table 1 diagnostics-15-00755-t001:** USS findings of abnormal lung pathology.

Pathology	Lung Ultrasound Pattern and Eponymous Signs
Normal aerated lung	Movement of the pleural line with tidal ventilation Presence of A-lines (Figure 1) Maximum of 2 B-lines per image
Cardiogenic pulmonary oedema	Homogenous B-line distribution >3 per image (Figure 2A) Regular thin pleura Possible pleural effusions
Interstitial lung disease	Irregular thickened pleura in moderate to severe disease Multiple diffuse bilateral B-lines
ARDS/ALI	Non-homogenous B-line distribution Irregular thickened pleura Sub-pleural consolidations
Pneumothorax	Absent lung sliding Stratosphere sign/barcode sign M-mode demonstrates only parallel horizontal lines indicating no aerated lung Lung pulse—absence of lung sliding with pulsed motion synchronous to heartbeat Lung Point—point at which pneumothorax meets with normal lung sliding
Pleural effusion	Interpleural hypo/anechoic space (Figure 3A,B) Jelly fish sign—lung moving within effusion appears jelly fish like
Consolidations	Shred sign—small sub-pleural consolidations (Figure 2B) Tissue like pattern/lung hepatisation—homogeneous texture of a lobe, similar to abdominal parenchyma Air bronchograms—hyperechoic branching structure within consolidation

**Table 2 diagnostics-15-00755-t002:** Quantitative lung ultrasound score qLUSS [40].

Score	qLUSS
Score 0—normal aeration	A-lines max 2 B-lines
Score 1—moderate loss of aeration	Artefacts occupying < 50% of the pleura
Score 2—severe loss of aeration	Artefacts occupying > 50% of the pleura
Score 3—complete loss of aeration	Tissue like pattern

## Data Availability

Not applicable.

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
