# Peer review of "Lung Ultrasound in Critical Care: A Narrative Review"

_diagnostics, 2025, doi:10.3390/diagnostics15060755_

Round 1
Reviewer 1 Report
Comments and Suggestions for Authors
This is a well-written narrative review article. The authors outline the current
uses of LUS within the context of published protocols, associated pathologies,
LUS scoring systems and their applications whilst exploring the more novel
uses. Although there have been several relevant (POCUS-lung) review
articles in recent literatures (see the following examples), it is always valuable
to discuss and educate such clinical experiences and pitfalls of using POCUS
in a variety of clinical scenarios. I have no major issues on this manuscript.
Minor: The reference-43 should be cited as follow:
1. Smit MR, Mayo PH, Mongodi S. Lung ultrasound for diagnosis and
management of ARDS. Intensive Care Med. 2024 Jul;50(7):1143-1145.
doi: 10.1007/s00134-024-07422-7.
2. Perhaps the authors could consider to share and add their own clinical
experiences/publication related to POCUS-lung in the text.
Examples of relevant literatures:
Beshara M, et al., MG. Nuts and bolts of lung ultrasound: utility, scanning
techniques, protocols, and findings in common pathologies. Crit Care.
2024 Oct 7;28(1):328. doi: 10.1186/s13054-024-05102-y.
Boumans MMA, et al., Diagnostic accuracy of lung ultrasound in
diagnosis of ARDS and identification of focal or non-focal ARDS
subphenotypes: a systematic review and meta-analysis. Crit Care. 2024
Jul 8;28(1):224. doi: 10.1186/s13054-024-04985-1.
Raheja R, et al., Application of Lung Ultrasound in Critical Care Setting: A
Review. Cureus. 2019 Jul 25;11(7):e5233. doi: 10.7759/cureus.5233..
Pietersen, P.I., et al. Lung ultrasound training: a systematic review of
published literature in clinical lung ultrasound training. Crit Ultrasound J
10, 23 (2018). https://doi.org/10.1186/s13089-018-0103-6
Murali A, et al., Lung ultrasound for evaluation of dyspnea: a pictorial
review. Acute Crit Care. 2022 Nov;37(4):502-515. doi:
10.4266/acc.2022.00780.
Heldeweg, M.L.A., et al., The impact of lung ultrasound on clinical-
decision making across departments: a systematic review. Ultrasound J
14, 5 (2022). https://doi.org/10.1186/s13089-021-00253-3
Gao X, et al., Application of POCUS in patients with COVID-19 for acute
respiratory distress syndrome management: a narrative review. BMC
Pulm Med. 2022 Feb 5;22(1):52. doi: 10.1186/s12890-022-01841-2.
Author Response
Many thanks for the reviewers helpful comments we have reviewed the points and have addressed the suggested changes as below with highlight to specific line for review
Point 1
Minor: The reference-43 should be cited as follow:
1. Smit MR, Mayo PH, Mongodi S. Lung ultrasound for diagnosis and management of ARDS. Intensive Care Med. 2024 Jul;50(7):1143-1145. doi: 10.1007/s00134-024-07422-7.
This has now been corrected on line 660
Point 2
Perhaps the authors could consider sharing and add their own clinical experiences/publication related to POCUS-lung in the text.
We agree as a narrative reviewer our own experience with lung ultrasound needed to be addressed and have added a sentence within the introduction. Line 59,60
Reviewer 2 Report
Comments and Suggestions for Authors
Dear authors,
Thank you for the chance to review your already very nicely written narrative review on lung ultrasound in critical care.
The text seems well researched, and will surely be handy for clinicians and students, but maybe also for paramedics.
I have three points that in my opinion need to be addressed:
-) Critical care is not only intensive care medicine or emergency department work, but also includes pre-hospital emergency medicine. Please include this aspect in your review, with special regard to respective literature, indications, various users of ultrasound in pre-hospital medicine ranging from paramedics to emergency physicians, and typical findings with respective therapeutic consequences. For instance also include ultrasound in cardiac arrest or in the evaluation of ventilation problems.
-) Your table listing pathologies and how they can be seen is already informative but can be further improved. You sometimes refer to figures, sometimes not, and the way of describing what you see is very heterogenous. Ideally, provide example pictures for each directly in the table.
-) There is no methodology section at all. While I know that this is a narrative review, at least some degree of methodology should be described. How did you come up with the literature you used? Was there a search strategy? Did you screen articles? Otherwise, your article is a mere book chapter and should then maybe be published as such.
Author Response
Many thanks for the reviewer comments, and we appreciate their time and expertise. Please see the responses to the reviewers comments.
Point 1
Critical care is not only intensive care medicine or emergency department work, but also includes pre-hospital emergency medicine. Please include this aspect in your review, with special regard to respective literature, indications, various users of ultrasound in pre-hospital medicine ranging from paramedics to emergency physicians, and typical findings with respective therapeutic consequences. For instance also include ultrasound in cardiac arrest or in the evaluation of ventilation problems.
We agree intensive care medicine is not solely provided within the confines of the intensive care unit and prehospital use of point-of-care ultrasound. Sections 2.2, 2.3 and 2.4 are not specific to intensive care and interpretation of images is generic across prehospitals or any hospital areas. We aimed for the narrative review to primarily be focused upon the secondary care intensive care settings as it is our area of expertise; however, we have made the following amendments on the following lines 15,16, 35,36 and added an additional section 7.4 lines 524-536 for prehospital consideration.
We have chosen to omit transthoracic echocardiography for cardiac assessment as the primary focus of the review was the use of lung ultrasounds.
We feel the area of prehospital lung ultrasound would also benefit from its own narrative review and expert authorship in the future.
Point 2
Your table listing pathologies and how they can be seen is already informative but can be further improved. You sometimes refer to figures, sometimes not, and the way of describing what you see is very heterogenous. Ideally, provide example pictures for each directly in the table
The table currently has listed the figures within the text, e.g., 2a. We do not feel that adding the figures within the table will add any additional clarity as they are already presented separately in a much better format. Moreover, incoprating the figures in a table will likely compromise the image quality.
Point 3
There is no methodology section at all. While I know that this is a narrative review, at least some degree of methodology should be described. How did you come up with the literature you used? Was there a search strategy? Did you screen articles? Otherwise, your article is a mere book chapter and should then maybe be published as such.
Thank you for the reviewers comments. For this narrative review, we did not employ a formalised or systematic search strategy. Instead, we adopted a targeted approach to identify and discuss relevant literature that aligns with the scope of this narrative review. The aim of this review was to provide an overview of the potential areas in which the LUS can be utilised in critical care settings, rather than providing all available literature of observational/RCT's, evaluating study quality, and their outcomes. Consequentially, our selection of resources was based on their relevance, significance, and contribution to this topic rather than an exhaustive search across multiple databases. Moreover, this approach allowed for flexibility in exploring emerging themes, ensuring a balanced representation of the literature that reflects the current understanding rather than the details of all studies published in this area.
However, we are currently considering undertaking systematic reviews within the areas discussed, comparing with other diagnostic methods, and providing a detailed overview of all relevant studies of specific lung conditions and the use of therapeutics such as mechanical ventilation and surfactant, which we feel would be best suited to a systematic methodology.
Round 2
Reviewer 2 Report
Comments and Suggestions for Authors
Dear authors,
Thank you for your reply. Unfortunately, you have failed to address any point of the ones I raised. Concerning prehospital lung ultrasound - there is not much literature on it, just add what is out there - this is some work but will improve your manuscript drastically. Concerning the table - I did not mean integrate the figures into the table only, but make is homogenous, meaning just refer to figures within the table. Concerning the methodology - I do not think that you just thought of literature that came to your mind, so I strongly suggest adding information describing the process of searching and adding literature. After all, this is - among other things - what differentiates an article in a peer-reviewed journal from a mere book chapter: the methodology must always be transparent.
Author Response
Dear Editor,
Thank you for the reviewers comments and your further clarification. Please see the responses to the comments below.
Point 1. Concerning prehospital lung ultrasound - there is not much literature on it, just add what is out there - this is some work but will improve your manuscript drastically.
Editorial Clarification: Prehospital Lung Ultrasound: The reviewer acknowledges your addition of this section but may expect a more thorough synthesis of existing literature. While you have incorporated key references, ensure the discussion clearly highlights the current evidence base, its limitations, and clinical implications
Many thanks for your guidance on this. While this was not the scope of this review (primarily aimed at critical care), we have considered this comment and introduced an additional section of the available literature (section 7.3). We have added additional references and expanded upon the sections 528-556 and 563-564.
The development of small handheld ultrasound devices has allowed the assessment of undifferentiated patients by prehospital and emergency care practitioners to improve their diagnostic accuracy [97]. Prehospital POCUS use has been shown to have particular benefit with cardiac arrest, to rule in or out cardiac activity , trauma to predict need for intervention such as pneumothorax or presence of free fluid within the abdomen and dyspnoea caused by congestive heart failure [98]. Considering point of care LUS alone in the assessment of undifferentiated patients with dyspnoea, paramedics experienced in the BLUE protocol and the BLUE profiles of pneumothorax , pneumonia and pulmonary oedema have shown a high diagnostic accuracy in prehospital setting in addition to aiding hospital diagnosis [99-101]. The recognition and treatment of acute heart failure by LUS in the prehospital setting has also been shown to improve paramedic diagnostic accuracy and such early recognition allows improved pre-hospital time to treatment [102]. Despite these potential benefits prehospital POCUS is yet to evidence mortality benefit to patients [103]. Concerns regarding prehospital interpretation of LUS images could be offset by real-time image review and supervision by trained POCUS clinicians in emergency medicine [104]. The introduction of AI models that could be translated and easily used prehospital may surpass the need for real-time supervision.
While the use of LUS in prehospital settings is increasing, similar to critical care, it is important to ensure appropriate standards and training are established to provide a safe and effective use of lung ultrasound in the prehospital setting [97, 98]. Currently within the UK, POCUS is being used by most prehospital services. However, a major barrier to the adoption of POCUS is the lack of governance and literature supporting its use in the prehospital setting despite the perceived benefits to patient care [103] Further prehospital LUS research should be based upon the diagnostic accuracy and associated time to treat patients, whilst the potential for AI-aided interpretation should be explored
Point 2. Concerning the table - I did not mean integrate the figures into the table only, but make is homogenous, meaning just refer to figures within the table.
Editorial Clarification: Table Formatting: The integration of figure references within the table appears appropriate. Please confirm that all citations align precisely with the figures in the main text and that formatting is consistent.
We have reviewed the table and believe it to be formatted correctly and aligns with the text, and figures are appropriately signposted within the table.
Point 3 Concerning the methodology - I do not think that you just thought of literature that came to your mind, so I strongly suggest adding information describing the process of searching and adding literature. After all, this is - among other things - what differentiates an article in a peer-reviewed journal from a mere book chapter: the methodology must always be transparent.
Editorial Clarification: Review Methodology: To enhance transparency, briefly describe your literature search strategy (e.g., databases, keywords) in the Methods section, even for a narrative review
We have now mentioned the review search methodology within lines 62-67.
Round 3
Reviewer 2 Report
Comments and Suggestions for Authors
Dear authors,
Thank you for responding to my comments. I feel that the manuscript is of sufficient quality for a publication to a broader audience now.